# Increased Lyso-Gb1 Levels in an Obese Splenectomized Gaucher Disease Type 1 Patient Treated with Eliglustat: Unacknowledged Poor Compliance or Underlying Factors

**DOI:** 10.3390/metabo15070427

**Published:** 2025-06-23

**Authors:** Evelina Maines, Roberto Franceschi, Giacomo Luppi, Giacomo Marchi, Giovanni Piccoli, Nicola Vitturi, Massimo Soffiati, Annalisa Campomori, Silvana Anna Maria Urru

**Affiliations:** 1Division of Pediatrics, S. Chiara General Hospital, APSS Trento, 38122 Trento, Italy; evelina.maines@apss.tn.it (E.M.); roberto.franceschi@apss.tn.it (R.F.);; 2Division of Radiology, S. Chiara General Hospital, APSS Trento, 38122 Trento, Italy; giacomo.luppi@apss.tn.it; 3Department of Medicine, University of Verona, 37129 Verona, Italy; giacomo.marchi@aovr.veneto.it; 4CIBIO, Department of Cellular, Computational and Integrative Biology, University of Trento, 37129 Trento, Italy; giovanni.piccoli@unitn.it; 5Division of Metabolic Diseases, Department of Medicine, Padova University Hospital, 35128 Padova, Italy; nicola.vitturi@aopd.veneto.it; 6Hospital Pharmacy Unit, S. Chiara General Hospital, APSS Trento, 38122 Trento, Italy; annalisa.campomori@apss.tn.it

**Keywords:** Gaucher disease, eliglustat, LysoGb1, splenectomy, obesity, case report

## Abstract

Eliglustat (Cerdelga^®^) is a potent and specific inhibitor of the enzyme glucosylceramide synthase and serves as a substrate reduction therapy for adult patients with Gaucher disease type 1 (GD1). It prevents the accumulation of several lipids, including glucosylsphingosine (also known as Lyso-Gb1). In addition to its role in diagnostics, Lyso-Gb1 has been proven to be a reliable biomarker for assessing disease severity and monitoring treatment efficacy. We present the case of an obese, splenectomized GD1 patient on long-term enzyme replacement therapy (ERT) who reported worsening fatigue and showed a progressive increase in Lyso-Gb1 levels after switching treatment from ERT to eliglustat. We provide a discussion of the potential clinical factors contributing to this outcome. As seen with ERT, Lyso-Gb1 levels during eliglustat treatment appear to respond earlier than other biochemical and clinical parameters. An increase in Lyso-Gb1 may signal early compromised clinical efficacy of the treatment. Data on biochemical and clinical outcomes in splenectomized or obese patients treated with eliglustat are limited, and the role of specific genotypes requires further clarification. The variability in responses to eliglustat highlights the complexity of GD and underscores the need for personalized approaches to treatment and monitoring.

## 1. Background

Eliglustat is a potent, specific inhibitor of the enzyme glucosylceramide synthase (GCS) and serves as a substrate reduction therapy (SRT) for adult patients with Gaucher disease type 1 (GD1). This SRT inhibits GCS and reduces the synthesis of glucosylceramide (also known as GL-1 or GB-1 or GlcCer)—the primary substrate of β-glucocerebrosidase (GCase)—ultimately preventing GL-1 accumulation and alleviating clinical manifestations. In addition to GL-1, several other lipids are accumulated in GD patients. One such lipid is the deacylated form of GL-1, glucosylsphingosine (also referred to as GlcSph, Lyso-GL1, or Lyso-Gb1) [1] (Figure 1).

Thanks to the availability of robust methods for its quantification in plasma, serum, or dried blood spot (DBS), Lyso-Gb1 has become an excellent diagnostic biomarker for GD, outperforming other markers such as chitotriosidase enzymatic activity (CHITO) and C-C Motif Chemokine Ligand 18 (CCL18) [2]. Beyond its role in diagnostic processes, Lyso-Gb1 has emerged as a reliable biomarker for evaluating disease severity and monitoring treatment efficacy [3].

Marked reductions in plasma Lyso-Gb1 levels were observed during the first year of eliglustat therapy in treatment-naïve GD1 patients, with reduced levels maintained over 4.5-year (the ENGAGE trial) [4] and 8-year [5] treatment periods. The median percent reduction in Lyso-Gb1 levels was 84% after 4.5 years [4] and 92% after 8 years of eliglustat therapy [5]. Similar trends in biomarker response were observed for CHITO, CCL18, and GL-1 [4,5].

Notably, decreased Lyso-Gb1 levels were correlated with improved clinical parameters of the spleen, liver, hemoglobin (Hb), and platelets (Plt) (all *p* < 0.05) [3]. Interestingly, a study involving 169 GD1 patients under treatment with the enzymatic replacement therapy (ERT, 155 cases) or with eliglustat (14 cases) reported that, after propensity score matching to obtain comparable groups of patients on ERT vs SRT, lyso-Gb1 levels were lower among patients receiving SRT by 113 ng/ml (95% CI: 136–90.3 ng/ml *p* < 0.001) [2]. Plasma Lyso-Gb1 levels correlated significantly with CHITO (r = 0.59; *p* < 0.001), as well as with indicators of the severity of visceral disease, such as splenic volume (r = 0.27, *p* = 0.003) and liver volume (r = 0.28, *p* < 0.001) [2]. In contrast, the authors did not find any correlation between Lyso-Gb1 levels and the indicators of skeletal disease severity, including bone marrow burden score (r = 0.12, *p* = 0.39), Hb (r = 0.07, *p* = 0.73), and Plt counts (r = 0.14, *p* = 0.07) [2].

Another study based on 38 GD1 adult patients on long-term ERT (mean 13.3 years) who switched to eliglustat (mean 3.1 years), showed a marked decrease in plasma Lyso-Gb1 from 63.7 ng/mL (95% CI, 37.6–89.8) to 26.1 ng/mL (95% CI, 15.7–36.6) (*p* < 0.0001). Serum Lyso-Gb1 reached nearly normal levels in 15 patients (with the reported normal plasma Lyso-Gb1 level being 3.3 ng/mL). Concomitantly, CHITO fell from 1136.6 nmoL/mL/h (95% CI, 144.7–2128.6) to 466.9 nmoL/mL/h (95% CI, 209.9–723.9). After switching from ERT to eliglustat, spleen volume decreased (*p* = 0.003) and Plt counts increased (*p* = 0.026) [6].

We present here the case of an obese, splenectomized GD1 patient on long-term ERT who reported worsening fatigue and showed a progressive increase in Lyso-Gb1 levels after switching treatment to eliglustat despite good compliance. We discuss the potential clinical contributors to this outcome.

## 2. Case Report

The patient is a 31-year-old woman of Macedonian origin. She was splenectomized at the age of 15 years after a 10-year medical history of anemia and thrombocytopenia, as well as massive splenomegaly. The diagnosis of GD1 was made based on spleen histological findings. The assessment of GCase activity in lymphocytes recorded a value of 3.1 nmol/mg/h (lab standard values for normal activity: 30.5 ± 17.9). Further tests showed significantly elevated plasma CHITO activity (20,040 nmol/h/mL; lab standard average values 25.75 ± 17). Lyso-Gb1 measurement was not available at the time of diagnosis. Three heterozygous variants in the *GBA1* gene were detected (c.882 T > G [p.H294Q] and c.1342G > C [p.D448H] in cis; c.1226 A > G2 [p.N409S] in trans).

Treatment with miglustat (NB-DNJ, Zavesca^®^, Actelion Pharmaceuticals Deutschland GmbH, Basler Strasse 63-65, Freiburg, Germany) was initiated at 200 mg/day and subsequently increased to 300 mg/day when she was 15.5 years old. The treatment was stopped after 3 years because of gastrointestinal side effects and suboptimal control of the disease.

At 18.5 years of age, she started ERT—Imiglucerase (Cerezyme^®^, Genzyme, Genzyme Ireland Limited IDA Industrial Park, Old Kilmeaden Road, Waterford, Ireland) (50 UI/kg/2wks because of disease burden). Overall, the treatment was well tolerated. The patient showed a clear improvement in clinical symptoms, and the hematological picture progressively normalized (Figure 2A,B). During the treatment, Lyso-Gb1 levels significantly lowered in plasma (minimum value 13.01 nmoL/L; normal lab standard values: 1.12–3) and reached the value 106 ng/mL (normal lab standard values: 0–14) in DBS (Figure 2C,D). CHITO decreased to 39 nmol/h/mL (normal lab standard values 8–121).

During the ERT treatment, the patient progressively gained 20 kg. She had two pregnancies. Nevertheless, her liver volume decreased (Figure 2F).

Genotyping for CYP2D6 revealed that she was an extensive metabolizer.

When the patient was 26.3 years old, the treatment was switched from ERT to eliglustat (84 mg BID) due to oral preference. At that time, she was obese, with a body weight of 119 Kg and a body mass index (BMI) of 40 kg/m^2^.

We observed a progressive increase in LysoGb1 levels in both plasma and DBS (Figure 2C,D). Hb and Plt levels remained in the normal range, but Plt levels dropped (Figure 2A,B). In particular, the variation in platelet count observed in our patient during eliglustat therapy remained within the 20% threshold, defined as stable disease in the ENCORE trial [7]. Her liver volume increased (Figure 2F).

The patient reported no overt adverse effects. She did not take other drugs. She reported good compliance and no treatment breaks. The hospital pharmacy confirmed the appropriate delivery of the drug.

After reporting fatigue worsening, the patient chose to return to ERT (50 UI/kg/2 wks of Imiglucerase). She reported an improvement in general well-being after 1–2 months of therapy, and her platelet count increased (Figure 2A). Lyso-Gb1 and CHITO values significantly decreased after only 3 months of ERT (Figure 2C–E).

## 3. Discussion

Our patient reported progressively worsening fatigue and showed a progressive increase in Lyso-Gb1 levels after switching treatment from ERT to eliglustat, despite good therapeutic adherence, the absence of treatment interruptions, and no potential drug–drug interactions.

It is known that non-adherence to medication is a problem that affects up to 50% of patients with chronic diseases. It may jeopardize the achievement of the therapeutic goals and increase the burden on the healthcare system [8]. The ELIPRO (ELIglustat Patient Reported Outcomes) study evaluated treatment adherence for eliglustat and reported treatment outcomes for over 1 year in GD1 patients, with a particular focus on patient-reported outcomes (PROs) [9]. A total of 60 patients were stratified by duration of eliglustat treatment, with 29 in Group 1 (treatment > 6 months) and 31 in Group 2 (treatment ≤ 6 months). Notably, 57 patients had previously received other treatments for GD1 (91% ERT). The adherence to eliglustat was measured by the eight-item Morisky Medication Adherence Scale (MMAS-8; scale of 0–8). At 6 months, 58% of Group 2 patients showed medium adherence (6 < MMAS-8 < 7.75), while 21% showed high adherence (MMAS-8: 8) (mean MMAS-8: 6.7 ± 1.0); similar results were obtained in Group 1 (50% showed high compliance, 35% showed medium compliance; mean MMAS-8: 6.8 ± 1.4).

Poor treatment adherence can cause a progressive increase in LysoGb1 levels and worsen clinical manifestations [1]. Nevertheless, in our patient, the hospital pharmacy confirmed regular drug dispensation, and the patient reported good adherence. Additionally, her medical history did not list treatment breaks.

The consequences of ERT treatment interruptions or dose reductions are well described. In particular, it is known that laboratory parameters respond earlier than clinical parameters [10,11,12]. In contrast, data on the clinical and biochemical consequences of eliglustat treatment interruptions are lacking. Gayed et al. [13] described a case involving a Caucasian male with type 3 Gaucher disease who exhibited poor adherence to treatment. He was treated with imiglucerase 35 U/kg every two weeks from the age of 22 months to the age of 4.9 years, then increased to 60 U/Kg every two weeks. Throughout the ERT treatment, the platelet counts consistently remained within the normal reference range, whereas CHITO activity was persistently elevated (average: 994.5 nmol/h/mL, range: 680–1852). At 18 years of age, the splenomegaly dropped to 2.2 multiples of normal (MN), and a pelvic MRI revealed bone involvement with a silent avascular necrosis of the femoral head. Due to significant needle phobia, the patient switched to eliglustat 84 mg BID at 18.2 years of age. Pelvic MRI at approximately 20 years old showed resolution of the avascular necrosis of the femoral head. The CHITO activity decreased from 1107 nmoL/h/mL to 467 nmoL/h/mL after 8 months of SRT. One year later, CHITO levels began to rise and eventually exceeded those measured during imiglucerase treatment (average: 3060 nmol/h/mL, range: 467–7515). Further evaluation revealed increased plasma Lyso-Gb1 levels (average: 350.3 nmoL/L, range: 66.1–747.2) and CHITO activity, as well as worsening of the splenic and hepatic volumes at age 25 years. The patient described by Gayed et al. [13] reported very poor adherence to the therapeutic regimen (missing one to three doses weekly).

Although our patient was compliant, two important clinical features must be considered—obesity and splenectomy.

The technical data sheet on eliglustat reports that gender, body weight, age, and race have limited or no impact on its pharmacokinetics. Nevertheless, the phase 1 study of eliglustat (Genz-112638)—which evaluated safety, tolerability, and pharmacokinetics—included only healthy volunteers weighing between 50 and 100 kg [14]. The phase 3 studies evaluating the efficacy and safety of eliglustat (Genz-112638) did not report the participants’ mean body weight or the weight-based distribution of the study sample [7,15,16]. In addition, other studies reporting the plasma Lyso-Gb1 levels during eliglustat treatment did not reveal the weight of the patients [2,17]. Only Kleytman et al. [6] reported—among the demographic characteristics of the patient cohort—the weight in kilograms at the time of visits (mean 78.7 kg, min 48.1–max 131.8 kg before switch/ERT and mean 80, min 49.9–max 120 kg after switch/SRT). Nevertheless, without additional information such as BMI, it is unclear how many patients were obese. Plasma Lyso-Gb1 levels in eliglustat cohort reduced from 63.7 ng/mL (95% CI, 37.6–89.8) to 26.1 ng/mL (95% CI, 15.7–36.6) (*p* < 0.0001). Nevertheless, some patients presented an increase in the Lyso-Gb1 values, as shown in the Figure 5 of that paper. The authors discussed the role of weight at the time of eliglustat treatment, but they did not evaluate the correlations between Lyso-Gb1 levels and the weight of the patients.

While the ERT dosage is defined in IU/kg, SRT dosage is determined primarily by CYP2D6 genotype rather than patient weight [18]. This approach simplifies treatment design but requires careful monitoring to ensure that therapeutic levels are maintained across diverse patient populations. Without accounting for the body weight, patients with higher or lower BMI might experience markedly different levels of drug exposure, which could impact efficacy or increase the risk of adverse effects [19].

It is known that body fat distribution can significantly affect the pharmacokinetics of drugs, influencing their absorption, distribution, metabolism, and elimination. Furthermore, obesity may impact the pharmacodynamics of the treatment through changes in inflammatory cytokine profiles or adipose tissue signaling [20,21].

Splenectomy should also be considered in our case. It is known that splenectomy may have a significant impact on the course of GD. The spleen may act as a reservoir for undegraded substrate. Its removal may cause the redistribution of this material to other macrophage-rich organs [22].

The clinical development program for eliglustat consisted of two controlled phase 3 studies (ENGAGE [15] and ENCORE [7]) and a phase 3b study (EDGE) [16]. The ENGAGE study excluded total or partial splenectomized GD1 patients. The ENCORE and EDGE studies excluded total or partial splenectomized patients within 2–3 years prior to the randomization. Only the ENCORE study included patients who underwent total or partial splenectomy > 3 years before randomization. Nevertheless, the study does not report LysoGb1 levels in this cohort. The study of Murugesan et al. [2] evaluated plasma Lyso-Gb1 levels in 155 GD1 patients on ERT and only 14 patients on eliglustat. Splenectomized patients were 20.7% in the ERT group and 17.7% in the SRT group. Lyso-Gb1 levels correlated with CHITO (r = 0.59 *p* < 0.001), CCL18 (r = 0.62 *p* < 0.001), hepatomegaly (r = 0.28 *p* < 0.001), splenomegaly (r = 0.27 *p* = 0.003), treatment mode (*p* < 0.001) and, interestingly, splenectomy (*p* = 0.01). Splenectomy was associated with higher serum Lyso-Gb1 levels compared to patients with intact spleens.

Other factors might underline the increase in Lyso-Gb1 levels, such as drug–drug interactions, CYP2D6 status and polymorphisms, genotypes, and epigenetic factors.

The concomitant administration of other drugs metabolized by CYP2D6, and, to a lesser extent, CYP3A may interfere with eliglustat metabolism [23]. However, our patient did not take any such medications. Furthermore, any drug that influences the metabolic pathways of sphingolipids could theoretically impact Lyso-Gb1 levels. For instance, medications affecting lysosomal function, glycosphingolipid metabolism, or autophagic flux could indeed alter Lyso-Gb1 levels and thus interfere with the monitoring of GD [24,25,26].

Despite well-established genotype–phenotype correlations [27,28], we are still unable to predict the clinical course of an individual GD1 patient at the time of diagnosis. Moreover, different genotypes respond differently to SRT compared to ERT [29]. It is possible that other genetic, epigenetic, or environmental factors may modulate the disease phenotype and response to treatment. Indeed, epigenetic mechanisms—including DNA methylation, histone modifications, and microRNA expression—can also influence the transcription of drug-metabolizing genes, such as *CYP2D6* [30].

The patient was identified as a CYP2D6 extensive metabolizer, a metabolic phenotype associated with lower plasma eliglustat concentrations compared to slower metabolizers [14]. In addition, it is known that plasma eliglustat concentration can vary even among individuals with the same CYP2D6 metabolizer status [31]. Therefore, sub-therapeutic drug levels due to an insufficient dose may also explain the patient’s clinical outcome.

Eliglustat is typically prescribed as a fixed dose of 84 mg once or twice daily for adults. This dosing strategy was chosen because, within the studied weight range, body weight did not substantially affect plasma drug levels [14]. However, this could not be applied to individuals with combined risk factors for reduced plasma concentrations, such as obesity and extensive metabolizer status.

As Lyso-Gb1 is considered a sensitive and early biomarker of substrate accumulation, its rise in this context—accompanied by a parallel increase in chitotriosidase—likely could reflect suboptimal drug exposure. These observations underline the potential value of individualized monitoring in selected patients.

Our patient carries a double mutation in cis, H294Q and D448H. This double mutation, previously annotated as H255Q and D409H, is mainly found in patients of Greek and Balkan origin [32]. Interestingly, the mutation P415R, located near the D409 residue, interferes with the binding to LIMP-2, the receptor involved in the trafficking of GCase from the Golgi to the lysosomes [33,34]. An intriguing hypothesis is that the D409H mutation similarly alters the trafficking of beta-glucocerebrosidase, potentially explaining the different Lyso-Gb1 levels observed in our patient during ERT (which provides the wild-type protein) versus eliglustat therapy.

To the best of our knowledge, no study has described Lyso-Gb1 levels in patients with the double D409H–H255Q mutation.

As often occurs with rare diseases, the number of eligible patients limits the power of clinical studies. Nevertheless, despite this limitation, certain patients should still be excluded.

Common reasons for exclusion include age, disease severity, or the presence of co-morbidities. These criteria aim to reduce variability and confounding factors, at the cost of a less representative patient population. The inclusion of certain patient subgroups in rare disease studies is critical but often challenging due to their unique health conditions or complications. This is particularly true for obese patients. It is difficult to apply findings derived from non-obese patients to obese individuals. Consequently, many clinical trials exclude obese patients, citing issues in dosage adjustments, the need for specialized care, or concerns about confounding variables in trial outcomes.

Splenectomized patients, on the other hand, face exclusion due to their altered immune function. This peculiar physiology may alter disease progression, especially in the case of GD1, thus complicating the interpretation of trial outcomes.

However, by excluding these subgroups, researchers may unintentionally limit the possibility to generalize their findings. Such an issue raises concern, especially in the context of rare diseases where each patient represents valuable data for understanding the condition [35,36].

## 4. Conclusions/Study Highlights

As reported for ERT, LysoGb1 during treatment with eliglustat seems to respond earlier than other biochemical and clinical parameters.

An increase in LysoGb1 levels may serve as an early indicator of reduced clinical efficacy of the treatment. Data on biochemical and clinical outcomes in splenectomized or obese patients treated with eliglustat are limited, and the role of specific genotypes, modifier genes, and epigenetic factors warrants further investigation. The variability in response to eliglustat underscores the complexity of GD and the need for personalized approaches.

## Figures and Tables

**Figure 1 metabolites-15-00427-f001:**
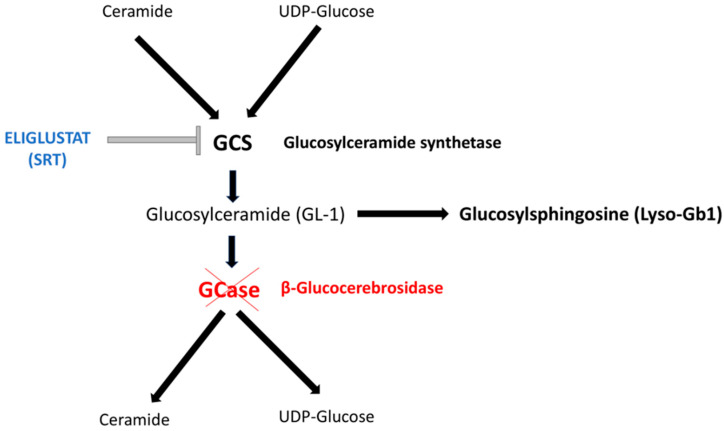
The scheme depicts the biochemical pathway of Gaucher Disease (GD). Enzymatic deficiency responsible for GD is marked in red. Eliglustat (marked in blue color) is a potent and specific inhibitor of the enzyme glucosylceramide synthetase (GCS) and acts as a substrate reduction therapy (SRT). SRT reduces the synthesis of glucosylceramide (also known as GL-1 or GB-1 or GlcCer)—the main substrate of β-glucocerebrosidase (GCase)—by inhibiting GCS and thus preventing the accumulation of GL-1. Beyond GL-1, several additional lipids accumulate in GD patients. One of these is the deacylated form of GL-1, glucosylsphingosine (also referred to as GlcSph, Lyso-GL1, or Lyso-Gb1), a reliable biomarker for evaluating disease severity and monitoring treatment efficacy.

**Figure 2 metabolites-15-00427-f002:**
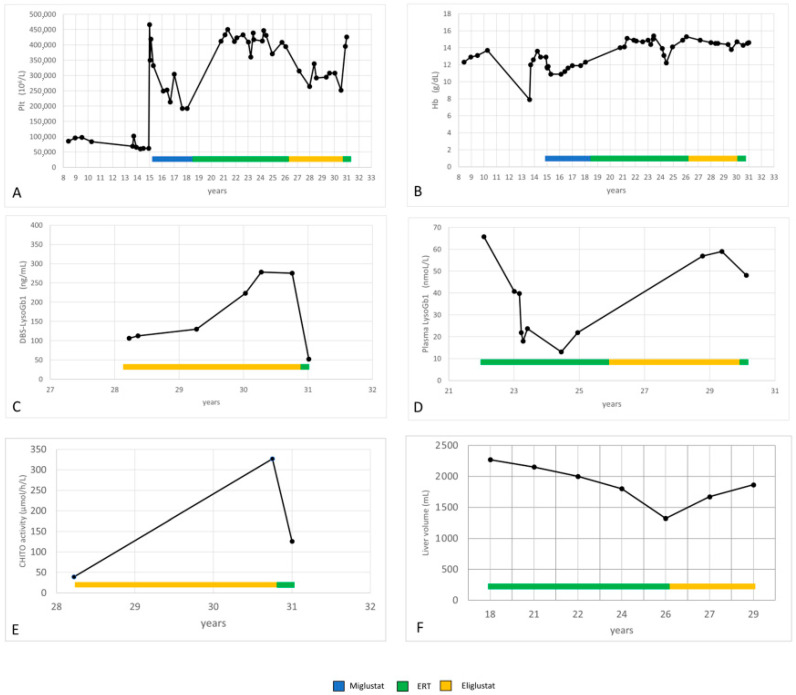
(**A**–**F**): Hemoglobin, platelets, LysoGb1, CHITO and liver volume values during miglustat, ERT and eliglustat treatment.

## Data Availability

The raw data supporting the conclusions of this article will be made available by the corresponding author on request.

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
