# Peer review of "Increased Lyso-Gb1 Levels in an Obese Splenectomized Gaucher Disease Type 1 Patient Treated with Eliglustat: Unacknowledged Poor Compliance or Underlying Factors"

_metabolites, 2025, doi:10.3390/metabo15070427_

Round 1

Reviewer 1 Report

Comments and Suggestions for Authors

This is a very interesting case report and will be hypothesis-generating, about a patient with Gaucher disease type 1 who had a suboptimal response to substrate reduction therapy possibly because of PK issues related to body size

I make the following specific comments and suggestions:

  1. Line 67, page 2: I do not know what is meant by "More in details" and these words should be removed.
  2. Line 69: "Plasmatic" should be replaced by "Plasma".
  3. Line 94 page 3: Lyso-Gb1 is a concentration not a dosage as stated.
  4. Line 102: This is an unusual dose or imiglucerase. I think a word or two of explanation as to why that was chosen would be helpful.
  5. Line 112: I would remove "median" as this is only relevant if you are referring to a weight variation over a specific period of time which would not be very helpful.
  6. Line 120: "3 months earlier" would be better that "3 months ago".
  7. Line 122; replace "platelets" with "platelet".
  8. Line 127, page 4. Correct the spelling of "platelets" in the figure legend.
  9. Line 130: What does "...worsening of her state" actually mean? In a case report, we need to know what it was that the patient was actually reporting (e.g. fatigue or bone pain or something else??). As far as I can see, the only objective measure of deterioration other than the Lyso-Gb1 was the platelet count.
  10. Line 219: Please define what you actually mean by "epigenetic factors" here and later in the Discussion.

There are two general points that should be discussed that will enhance the paper.

  1. The platelet count variation seen in this patient are within the definition of stability in the ENCORE study where deterioration of 20% of the platelet count was considered stable disease and this is in the range of platelet count decrease that the described patient experienced.
  2. Figure 1 clearly shows that Lyso-Gb1 should be significantly decreased by effective SRT independent of any assessment of disease activity. The authors have shown, in this case, a parallel change of chitotriosidase and I feel that this is an important matter to discuss and would agree that in this case, the rise in Lyso-Gb1 is probably due to pharmacokinetic issues possibly related to the patient's high BMI.
Comments on the Quality of English Language

English is somewhat flawed in many places, most of which I have outlined in the comments to authors.

Author Response

This is a very interesting case report and will be hypothesis-generating, about a patient with Gaucher disease type 1 who had a suboptimal response to substrate reduction therapy possibly because of PK issues related to body size.

We are very grateful for the reviewer’s insightful comments. The suggestions have significantly contributed to improving the quality of our work. 

All revisions are in red and underlined.

I make the following specific comments and suggestions:

  1. Line 67, page 2: I do not know what is meant by "More in details" and these words should be removed.

Answer: “more in details” has been removed, as suggested.

  1. Line 69: "Plasmatic" should be replaced by "Plasma".

Answer: “plasmatic” has been changed in plasma, as suggested.

  1. Line 94 page 3: Lyso-Gb1 is a concentration not a dosage as stated.

Answer: we apologize for the mistake. We have removed “dosage” 

  1. Line 102: This is an unusual dose or imiglucerase. I think a word or two of explanation as to why that was chosen would be helpful.

Answer: we use a dosage of 50 UI/Kg because of the disease burden of the patient.

  1. Line 112: I would remove "median" as this is only relevant if you are referring to a weight variation over a specific period of time which would not be very helpful.

Answer: thank you for the suggestion.

  1. Line 120: "3 months earlier" would be better that "3 months ago".

Answer: thank you for the suggestion.

  1. Line 122; replace "platelets" with "platelet".

Answer: thank you for the suggestion.

  1. Line 127, page 4. Correct the spelling of "platelets" in the figure legend.

Answer: thank you for the suggestion.

  1. Line 219: Please define what you actually mean by "epigenetic factors" here and later in the Discussion.

We thank the reviewer for this comment. We added a brief description of epigenetic factors in the discussion section (see lines 236-239)

There are two general points that should be discussed that will enhance the paper.

  1. The platelet count variation seen in this patient is within the definition of stability in the ENCORE study where deterioration of 20% of the platelet count was considered stable disease and this is in the range of platelet count decrease that the described patient experienced.

We thank the reviewer for this constructive comment. We have accordingly added a brief paragraph  in the case report to address the definition of stability in platelet counts based on the ENCORE study criteria, noting that the degree of variation observed in our patient falls within this accepted threshold (see lines 119-121)

  1. Figure 1 clearly shows that Lyso-Gb1 should be significantly decreased by effective SRT independent of any assessment of disease activity. The authors have shown, in this case, a parallel change of chitotriosidase and I feel that this is an important matter to discuss and would agree that in this case, the rise in Lyso-Gb1 is probably due to pharmacokinetic issues possibly related to the patient's high BMI.

We thank the reviewer for its suggestion. We included a brief paragraph in the discussion section (see lines 251-254).

Comments on the Quality of English Language

English is somewhat flawed in many places, most of which I have outlined in the comments to authors.

We thank the reviewer for his/her careful reading of the manuscript. We hope the revised version meets the expected standards of written English.

Reviewer 2 Report

Comments and Suggestions for Authors

This is an interesting article regarding an adult splenectomized female with Gaucher disease type 1 who showed a sub-optimal therapeutic response to eliglustat after transitioning from imiglucerase, followed by a renewed therapeutic response after resuming imiglucerase.  The data are well-presented and effectively utilize plasma lyso-Gb1 levels to monitor the biochemical response to treatment.  The authors raise potential explanations for the sub-therapeutic response (non-compliance, drug interaction), but they have omitted an important one - an inadequate dose leading to a sub-therapeutic drug level.  The patient was identified as a CYP2D6 extensive metabolizer, which is associated with lower plasma eliglustat levels than slower metabolizing variants (a figure is shown in the Peterschmitt et al J Clin Pharm publication from 2011).  There can be a range of plasma eluglustat levels even within individuals with the same CYP2D6 metabolizer level. While CYP2D6 is the predominant route of metabolism for eliglustat, CYP3A4 also contributes, but to a lesser extent.  If the authors have archived samples, Sanofi may be willing to test the plasma eliglustat levels, which would provide definitive evidence of whether the sub-optimal treatment response was related to sub-therapeutic plasma eliglustat levels.  The authors should mention the possibility of a sub-therapeutic drug level in an obese, extensive metabolizer in the discussion.  Without actual plasma eliglustat levels, the authors could in the future increase the dose and/or frequency of eliglustat and see whether the lyo-GL1 level decreases (I would monitor the ECG at 2 hr post-dose for any clinically significant changes in the PR and QTc intervals and ask Sanofi to measure the plasma levels at the same time).  As for this manuscript, the authors should at least mention the possibility of a sub-therapeutic drug level as the cause of the sub-therapeutic response and ask Sanofi whether it is possible to test peak and trough plasma eliglustat levels in exceptional cases like this.  Testing plasma eliglustat levels in patients with a sub-therapeutic response is an important point to bring up that has broad consequences for patients who would like to continue to receive an oral therapy. As the authors point out, the 84 mg once or twice daily dose is a one-size fits-all dose for adults.  The flat dose was chosen because the effect of weight within the range studied did not have a major effect on the plasma concentration.   This may not be the case in someone with two risk factors for having a lower plasma eliglustat level (obesity and extensive metabolizer).

Author Response

This is an interesting article regarding an adult splenectomized female with Gaucher disease type 1 who showed a sub-optimal therapeutic response to eliglustat after transitioning from imiglucerase, followed by a renewed therapeutic response after resuming imiglucerase.  The data are well-presented and effectively utilize plasma lyso-Gb1 levels to monitor the biochemical response to treatment.  The authors raise potential explanations for the sub-therapeutic response (non-compliance, drug interaction), but they have omitted an important one - an inadequate dose leading to a sub-therapeutic drug level.  The patient was identified as a CYP2D6 extensive metabolizer, which is associated with lower plasma eliglustat levels than slower metabolizing variants (a figure is shown in the Peterschmitt et al J Clin Pharm publication from 2011).  There can be a range of plasma eliglustat levels even within individuals with the same CYP2D6 metabolizer level. While CYP2D6 is the predominant route of metabolism for eliglustat, CYP3A4 also contributes, but to a lesser extent.  If the authors have archived samples, Sanofi may be willing to test the plasma eliglustat levels, which would provide definitive evidence of whether the sub-optimal treatment response was related to sub-therapeutic plasma eliglustat levels.  The authors should mention the possibility of a sub-therapeutic drug level in an obese, extensive metabolizer in the discussion.  Without actual plasma eliglustat levels, the authors could in the future increase the dose and/or frequency of eliglustat and see whether the lyo-GL1 level decreases (I would monitor the ECG at 2 hr post-dose for any clinically significant changes in the PR and QTc intervals and ask Sanofi to measure the plasma levels at the same time).  As for this manuscript, the authors should at least mention the possibility of a sub-therapeutic drug level as the cause of the sub-therapeutic response and ask Sanofi whether it is possible to test peak and trough plasma eliglustat levels in exceptional cases like this.  Testing plasma eliglustat levels in patients with a sub-therapeutic response is an important point to bring up that has broad consequences for patients who would like to continue to receive an oral therapy. As the authors point out, the 84 mg once or twice daily dose is a one-size fits-all dose for adults.  The flat dose was chosen because the effect of weight within the range studied did not have a major effect on the plasma concentration.   This may not be the case in someone with two risk factors for having a lower plasma eliglustat level (obesity and extensive metabolizer).

We are very grateful for the reviewer’s insightful comments. The suggestions have significantly contributed to improving the quality of our work. 

All revisions are in red and underlined.

We included a paragraph in the Discussion section (lines 240–250) addressing the possibility of a sub-therapeutic drug level in an obese patient who is an extensive metabolizer. 

We contacted Sanofi, who stated that it is not possible to test plasma eliglustat levels in their laboratories. 

Unfortunately, no plasma samples from the patient are available.

Round 2

Reviewer 1 Report

Comments and Suggestions for Authors

I thank the authors for improving their paper by taking into account specific suggestions.